# An Unusual and Exaggerated Local Cutaneous Reaction Following Re-Irradiation and Fulvestrant Administration: A Clinical Conundrum

**DOI:** 10.3390/diagnostics15081017

**Published:** 2025-04-16

**Authors:** Valentina Zagardo, Dorotea Sciacca, Gianluca Ferini

**Affiliations:** 1Department of Radiation Oncology, REM Radioterapia Srl, 95029 Viagrande, Italy; valentina.zagardo@grupposamed.com; 2Mediterranean Institute of Oncology, 95029 Viagrande, Italy; dorotea.sciacca@grupposamed.com; 3Department of Medicine and Surgery, University Kore of Enna, 94100 Enna, Italy

**Keywords:** radiation dermatitis, Fulvestrant, radiation recall, breast cancer, radiotherapy, bone metastasis

## Abstract

A 56-year-old female with a history of Luminal A breast cancer, previously treated with surgery, radiotherapy, and systemic therapy, underwent palliative re-irradiation in November 2024 for painful bone metastases. Three weeks later, following the initiation of Fulvestrant, she developed a grade 3 erythematous reaction localized to the re-irradiated area. The reaction persisted with minimal improvement over two months, despite symptomatic management. No infectious or allergic etiologies were identified, and dosimetric analysis confirmed that the delivered radiation dose to the skin was insufficient to directly induce such a reaction. Notably, the erythema was most pronounced along a pre-existing surgical scar, suggesting a localized inflammatory response. Given the temporal relationship with Fulvestrant administration, we hypothesize a drug-induced recall-like phenomenon, though no previous reports have specifically linked Fulvestrant to such an event. This case underscores the need for awareness of unexpected cutaneous reactions following re-irradiation and highlights the potential role of systemic therapies in modulating local tissue responses.

**Figure 1 diagnostics-15-01017-f001:**
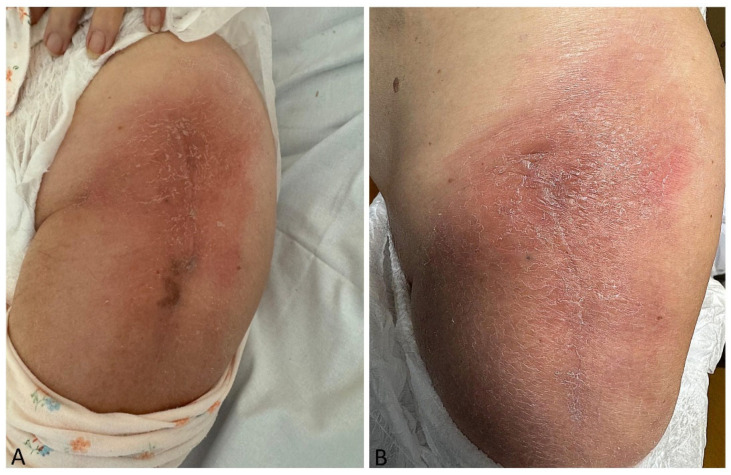
Severe radiation dermatitis following re-irradiation for bone metastases and Fulvestrant administration. A 56-year-old female patient with a history of Luminal A breast cancer, previously treated with hormonal therapy, surgery, and radiotherapy, was referred to the radiotherapy department in March 2021 following the stabilization of a pathological fracture of the left hip. Her medical history was unremarkable, except for hypertension and hypercholesterolemia, for which she was receiving appropriate treatment. On physical examination, the patient reported moderate pain localized to the pelvis and lumbar vertebrae. Computed tomography (CT) imaging revealed predominantly lytic lesions consistent with widespread bone disease. Given these findings, the patient underwent palliative radiotherapy using a half-body irradiation (HBI) technique, targeting the entire pelvic bone and lumbar vertebrae, with a total dose of 30 Gy delivered in 10 fractions [1]. No acute or late toxicities were documented. The patient experienced clinical benefit from the treatment and continued systemic therapy, maintaining disease stability until October 2024, when she developed severe pain in the left ischial branch, with further disease progression documented on CT at this site. At that time, carboplatin-based chemotherapy was discontinued due to both disease progression and poor tolerance, including febrile neutropenia. In November 2024, the patient underwent a radiotherapy consultation. Given the clinical scenario (severe pain poorly responsive to pharmacological therapy) and the interval since the previous irradiation (three years), re-irradiation to the painful site was indicated. A total dose of 20 Gy in five fractions was delivered between 3 and 7 November. On 23 November, the patient initiated a new systemic therapy with Fulvestrant. In December 2024, she presented to her general practitioner with intense erythema along the ventral portion of the thigh, classified as grade 3 (G3) by the physician. She was prescribed moisturizers, emollients, and topical antibiotics, with mild symptomatic relief. On 9 January, the patient was hospitalized at our institution due to hypocalcemia. On physical examination, the erythema on the ventral thigh, corresponding to the re-irradiated area, was resolving (the box on the left (Figure 1A) (**A**) shows the clinical picture 1 month after its onset, while the one on the right (Figure 1B) (**B**) shows the re-evaluation at 2 months, with no significant changes). No induration or tenderness was noted, and the patient denied any trauma to the site. She also denied the use of new topical agents that could suggest contact dermatitis. We hypothesized that this phenomenon was consistent with “radiation recall”, an inflammatory tissue reaction “recalled” following the introduction of a new systemic therapy [2,3]. This phenomenon may be induced by oncological agents as well as by other medications, including anti-inflammatory drugs [4,5,6]. In the present case, the patient had not received any new treatments in recent months apart from Fulvestrant, which was therefore considered the potential trigger. While previous reports have described hormone therapy agents, such as letrozole [7] and exemestane [8], and CDK inhibitors [9,10] as possible inducers, no cases specifically associated with Fulvestrant have been documented in the literature. Laboratory tests were conducted to rule out infectious causes and exclude alternative etiologies. Additionally, we excluded the possibility of a rare allergic reaction to Fulvestrant, as described with other similar agents [11,12], given the localized nature of the erythema, the absence of pruritus, the lack of pustular lesions, and the fact that continued drug administration did not exacerbate the condition. We subsequently re-evaluated the delivered treatment plans (Figure 2A,B). The analysis of re-irradiation planning demonstrated that only a total dose of 5 Gy had reached the skin, an insufficient dose to directly induce this reaction, whereas the initial irradiation had delivered a total of 10 Gy to the skin. Furthermore, the potential for the back-scattering of the radiation beam due to the hip prosthesis was considered negligible, as the prosthesis was deeply seated and at a considerable distance from the affected skin. Notably, the cutaneous reaction was localized to the site of the most recent radiotherapy but was more pronounced along the surgical scar from the prior fracture stabilization procedure. This observation raises the hypothesis that Fulvestrant may have triggered the radiation recall phenomenon by preferentially promoting the deposition of inflammatory cells along the surgical scar. As no established explanation for this phenomenon currently exists in the literature, we can only speculate, as is the case for other unexplained radiation-related phenomena [13,14]. Fulvestrant was not discontinued, as no alternative therapeutic options were available for the patient, given that this was her third-line treatment. At the time of writing, approximately two months after onset, the cutaneous reaction had shown only slight improvement and has since remained stable (Figure 1B). Unfortunately, a biopsy was not performed due to the patient’s refusal. Therefore, while we cannot provide a definitive diagnosis, we aim to alert clinicians to this complex clinical scenario, which may be a consequence of Fulvestrant administration and could warrant treatment discontinuation if it progresses to a more severe form.

**Figure 2 diagnostics-15-01017-f002:**
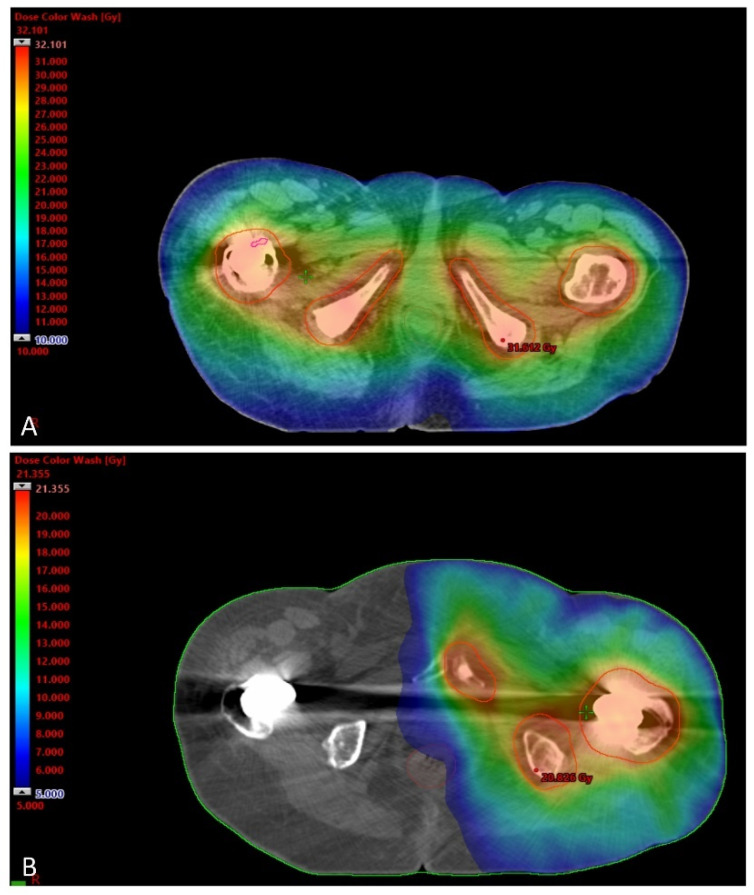
(**A**) Initial irradiation treatment plan, showing the 10 Gy isodose reaching the skin. (**B**) Re-irradiation plan, demonstrating a total skin dose of 5 Gy. The red line delineates the planning target volume.

## Data Availability

No new data were created or analyzed in this study.

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
