# Peer review of "An Unusual and Exaggerated Local Cutaneous Reaction Following Re-Irradiation and Fulvestrant Administration: A Clinical Conundrum"

_diagnostics, 2025, doi:10.3390/diagnostics15081017_

Round 1

Reviewer 1 Report

Comments and Suggestions for Authors

Dear authors,

I did find this case report article very interesting, accordingly I insist on an inquiry of patient drug history more precisely, especially about any comorbidities and probable medicine prescription.

Thank you.

Author Response

Reviewer 1:

Dear authors,

I did find this case report article very interesting, accordingly I insist on an inquiry of patient drug history more precisely, especially about any comorbidities and probable medicine prescription.

Thank you.

Answer: We would like to thank the reviewer for the insightful comment. The patient had been suffering from hypertension and hypercholesterolemia for several years, for which she was regularly undergoing treatment. We have included this information in the manuscript. “Her medical history was unremarkable, except for hypertension and hypercholesterolemia, for which she was receiving appropriate treatment.”

We thank the reviewer for their comment, which has allowed us to improve the quality of the manuscript.

Reviewer 2 Report

Comments and Suggestions for Authors

Dear Authors,

I have read an article about an unusual local cutaneous reaction following re-irradiation. While interesting, I have several inputs for this manuscript:

1) Please separate Figure 1's caption with the rest of the paragraph. The authors are making it seem like Figure 1's caption is a part of the whole paragraph

2) The authors mention that the patient suffers from moderate pain (NRS 6) but NRS 7 is severe. While I understand that this may be the cut-off used, do consider omitting the use of NRS since it will just contradict your sentence

3) Why was citation [1] included in the case description? Was this patient already described in a previous case? Please elaborate.

4) Figure 1 seems to be the left hip, not the right. The re-irradiation planning seems to be targeting the left hip as well.

Author Response

Reviewer 2:

Dear Authors,

I have read an article about an unusual local cutaneous reaction following re-irradiation. While interesting, I have several inputs for this manuscript:

  • Please separate Figure 1's caption with the rest of the paragraph. The authors are making it seem like Figure 1's caption is a part of the whole paragraph

Answer: I would like to thank the reviewer for his comment; we have followed the suggestion provided.

  • The authors mention that the patient suffers from moderate pain (NRS 6) but NRS 7 is severe. While I understand that this may be the cut-off used, do consider omitting the use of NRS since it will just contradict your sentence

Answer: We would like to thank the reviewer for the insightful comment. As mentioned by the reviewer, the cutoff for severe pain is 7. We defined the pain as moderate because the patient reported a pain intensity of 6. We acknowledge that this is at the threshold between moderate and severe pain, and in order to avoid any contradictions, as suggested by the reviewer, we have refrained from using the NRS scale.

  • Why was citation [1] included in the case description? Was this patient already described in a previous case? Please elaborate.

Answer: We deleted such a reference as it is not strictly relevant.

  • Figure 1 seems to be the left hip, not the right. The re-irradiation planning seems to be targeting the left hip as well.

Answer: We would like to thank the reviewer for the comment and we apologize for the error. We regret the oversight; as pointed out by the reviewer, it pertains to the left hip, and we have corrected the mistake made.

We thank the reviewer for their comment, which has allowed us to improve the quality of the manuscript.

Reviewer 3 Report

Comments and Suggestions for Authors

The authors offer an intriguing clinical case that includes clinical evaluation, computed tomography, and a comprehensive follow-up. The figures and captions provide valuable insights into the consequence of fulvestrant administration combined with irradiation therapy, and could warrant treatment discontinuation.

Author Response

Reviewer 3:

The authors offer an intriguing clinical case that includes clinical evaluation, computed tomography, and a comprehensive follow-up. The figures and captions provide valuable insights into the consequence of fulvestrant administration combined with irradiation therapy, and could warrant treatment discontinuation.

Answer: We thank the reviewer for his comment. The reviewer has correctly identified the reason why we aimed to alert the scientific community to this potential adverse reaction, albeit a rare one. Indeed, as the reviewer has also emphasized, if it progresses to a severe form, it may necessitate treatment discontinuation.

Reviewer 4 Report

Comments and Suggestions for Authors

The present case report is interesting and provides useful information to the oncological community.

The following comments might improve the message:

One. The redaction is odd: A legend for Fig 1 is needed, apart from the text.

Two. Were laboratory tests done at onset of the erythema?

Three. In view of the literature (refs 4 & 5), Fulvestrant, a selective estrogen receptor degrader (SERD)  is a valuable candidate co-trigger of the radiation-induced cutaneous reaction. However, next to oncological medication all other medication, canonical and alternative, should be considered:  NSAIDs, (Zhang et al. An Bras Dermatol. 2023, 21: 1405)? Corticoids? Vaccination ? It is unlikely that the patient, suffering from advanced cancer, did not take any of these, or similar, drugs.

Author Response

Reviewer 4:

The present case report is interesting and provides useful information to the oncological community.

The following comments might improve the message:

One. The redaction is odd: A legend for Fig 1 is needed, apart from the text.

Answer: We thank the reviewer for the comment. As suggested, we have added the legend for Figure 1. “Radiation dermatitis following re-irradiation and Fulvestrant administration. On the left (A), the erythema is shown one month after its onset. On the right (B), the re-evaluation at two months demonstrates substantial stability of the condition”

Two. Were laboratory tests done at onset of the erythema?

Answer: We thank the reviewer for the comment. As stated in the manuscript, the laboratory tests were conducted to exclude possible infectious diseases.

Three. In view of the literature (refs 4 & 5), Fulvestrant, a selective estrogen receptor degrader (SERD)  is a valuable candidate co-trigger of the radiation-induced cutaneous reaction. However, next to oncological medication all other medication, canonical and alternative, should be considered:  NSAIDs, (Zhang et al. An Bras Dermatol. 2023, 21: 1405)? Corticoids? Vaccination ? It is unlikely that the patient, suffering from advanced cancer, did not take any of these, or similar, drugs.

Answer: We would like to thank the reviewer for the insightful comment.  Our patient had not received any vaccinations nor initiated any new pharmacological treatments other than fulvestrant in the recent period. However, as suggested by the reviewer, we have included in the discussion section the possibility that this phenomenon may also be triggered by non-oncological drugs. “This phenomenon may be induced by oncological agents as well as by other medications, including anti-inflammatory drugs [4]. In the present case, the patient had not received any new treatments in recent months apart from fulvestrant, which was therefore considered the potential trigger.

Round 2

Reviewer 2 Report

Comments and Suggestions for Authors

The authors have satisfactorily improved on all the requested points.